# Production and Shielding Effectiveness Features of Chopped Strands Backed-GdMnO_3_ Composites for 6.5–17.5 GHz Applications

**DOI:** 10.3390/ma18040863

**Published:** 2025-02-16

**Authors:** Mehriban Emek, Ethem İlhan Şahin, Jamal Eldin F.M. Ibrahim

**Affiliations:** 1Department of Physics, Golbasi Vocational School, Computer Technologies, Adiyaman University, Adıyaman 02040, Turkey; memek@adiyaman.edu.tr; 2Advanced Technology Research and Application Center, Adana Alparslan Türkeş Science and Technology University, Adana 01250, Turkey; eisahin@atu.edu.tr; 3Institute of Ceramics and Polymer Engineering, University of Miskolc, H-3515 Miskolc, Hungary; 4Department of Materials Technology, University of Bahri, Khartoum 12217, Sudan

**Keywords:** shielding effect, GdMnO_3_, mixed oxide, chopped strands, matrix composites

## Abstract

This research investigates the synthesis and characterization of GdMnO_3_/chopped strands composites using the conventional oxide mixing technique. A single-phase GdMnO_3_ compound was successfully formed through sintering at 1350 °C for 20 h. Structural analysis using X-ray diffraction (XRD), scanning electron microscopy (SEM), and energy-dispersive X-ray spectroscopy (EDS) confirmed phase purity and uniform grain morphology. The microwave shielding effectiveness of GdMnO_3_/chopped strands composites was evaluated within the 6.5–17.5 GHz frequency range using a network analyzer (NA). The GdMnO_3_/chopped strands composite with a 60–40% weight ratio exhibited superior shielding performance, achieving a minimum shielding effectiveness of −35.61 dB at 6.9 GHz, while the 80–20% composite reached −32.54 dB at 16.74 GHz. Both compositions demonstrated shielding effect values below −10 dB across wide frequency bands, with significant attenuation below −20 dB at various GHz ranges. The study demonstrates that by adjusting the content of the components in the samples, the microwave shielding effect performance of the GdMnO_3_/chopped strands composites can be easily controlled to meet the requirements of specific frequency bands. These findings highlight the potential of GdMnO_3_-based composites for tailored microwave shielding applications, particularly in the military, aerospace, and telecommunication industries.

## 1. Introduction

Electromagnetic waves within the GHz frequency range find extensive application in communication systems and a wide array of electrical and electronic devices. However, the extensive utilization of these waves in our daily lives, communication systems, and industrial settings results in the generation of significant amounts of undesirable waves. These waves, when present in the atmosphere, contribute to the proliferation of electromagnetic pollution commonly known as electromagnetic interference (EMI) [1].

In recent times, there has been a notable surge in the investigation of electromagnetic interference (EMI) primarily driven by the advancements in devices that operate within the gigahertz range. Notable examples include satellites, mobile phones, and cordless devices, which have contributed to the need for more comprehensive studies on EMI-related issues [2,3]. In our modern lives, we are continuously enveloped by a vast network of wireless microwave signals that remain invisible to our eyes. Despite their imperceptible nature, concerns have arisen regarding their potential impact on the environment we inhabit. Long-term exposure to silent and invisible high-intensity electromagnetic waves can cause physiological changes and health problems, as well as affecting genetic function and immune function [4,5]. Beyond these biological effects, electromagnetic susceptibility (EMS) to intentional electromagnetic interference (IEMI) presents a critical challenge, particularly in power electronics and secure communication systems. IEMI threats, categorized into conducted and radiated susceptibility, can severely disrupt or damage electronic systems through high-power electromagnetic pulses [6].

In order to prevent and reduce electromagnetic wave exposure, there are engineering techniques such as using personal protective equipment, limiting exposure time, and reducing exposure levels to acceptable levels [5]. In order to protect systems and people from this electromagnetic pollution, it is necessary to use shielding materials that shield against EMI or provide reflection loss. EMI shields can provide protection from electromagnetic waves by shielding unwanted EM waves, using mechanisms that cause them to be absorbed or reflected by the material [6,7,8]. For this reason, scientists and researchers are working on microwave absorbing and electromagnetic (EM) shielding materials and trying to improve existing ones [9].

Every year, there is a growing focus on the creation and advancement of novel composite materials for electromagnetic (EM) shielding and absorption purposes [10,11,12].

These materials find applications in various sectors, including both military and civilian domains [10]. EM shielding and absorbing materials play a crucial role in the functionality of portable electronics, wireless communication systems, aerospace technologies, military equipment, and medical devices. Their significance lies in ensuring the proper operation and performance of these various equipment and systems [12]. EMI shielding effectiveness relies on various contributing factors, and one of these crucial aspects is reflection loss. Reflection loss is contingent upon the interaction between mobile charge carriers, whether they are electrons or holes, and the incident electromagnetic wave. Another significant factor is absorption loss, which is influenced by the interaction between magnetic and electrical dipoles with EM waves [13]. Lastly, the multi-reflection effect plays a key role, referring to internal reflections occurring within the shielding material, preventing the passage of electromagnetic waves. This effect typically becomes more pronounced in scenarios where there are numerous and extensive surface or interfacial regions [14]. However, with the rapid advancement of contemporary electronic technology and sophisticated reconnaissance systems, the capabilities of detectors to identify and monitor targets have significantly improved [15,16]. Therefore, enhancing the target’s survivability hinges on impeding the enemy’s ability to detect, identify, track, and launch an attack. To achieve this, a specialized approach involves incorporating unique outer structures or surface coatings that render the target invisible to the enemy’s detection systems, thereby reducing its detectable signal characteristics [13,17,18,19]. To illustrate this, specific conditions must be met to avoid detection on radar. It is crucial that the radar signal directed towards an object is not reflected back in the same direction and does not emit any transmission simultaneously [13,20]. At present, radar and infrared sensing make up 60% and 30% of sensing technologies, respectively. As a result, a significant portion of research has been dedicated to achieving stealth capabilities in both radar and infrared domains [21,22]. To create an effective microwave absorber, it must satisfy two essential requirements known as impedance matching and wave attenuation [23,24,25]. Impedance matching ensures that incoming electromagnetic (EM) radiations can enter the material without being reflected by its surface. On the other hand, wave attenuation refers to the ability of the material to rapidly diminish EM waves that penetrate into it. Single-component materials often exhibit an inadequate electromagnetic wave absorption performance due to the difficulty in achieving both good impedances matching and a strong loss capacity simultaneously [26]. In addition, ferrite and other traditional microwave absorbers have a restricted range of applications due to their size, sensitivity to corrosion, high density, and expensive processing [27,28].

Electromagnetic shielding can also be defined as the process of reflecting the absorption and scattering of electromagnetic waves without entering or leaving a certain area. To prevent and reduce electromagnetic wave exposure, and to protect systems and people from this electromagnetic pollution, it is necessary to use shielding materials against EMI [29]. EMI shielding materials provide defense against electromagnetic waves through mechanisms designed to absorb or reflect unwanted EM waves [30]. From an environmental perspective, it is worth noting that the absorption mechanism is preferred over reflection [8,29,31]. EMI shielding is affected by multiple factors, including reflection loss. This is based on the interaction between mobile charge carriers (electrons or holes) and incoming EM waves. Another influential factor is absorption loss, which is determined by how magnetic and electric dipoles interact with EM waves. The third mechanism, called the multiple reflection effect, comes into play when there are large surface or interface areas within the shielding material, which leads to internal reflections [29]. EMI shielding uses composite materials containing particles, metal flakes, and discontinuous conductive additives such as carbon fibers and metallic wires; polymer matrix composites are preferred due to their versatility and ability to overcome material deficiencies [29,32].

Based on their chemical composition and crystallographic orientation, single-phase multiferroic materials may be divided into a number of distinct categories. These types include perovskite, rare earth manganite, REMnO_3_ (RE = Ho, Lu, Y, Gd and Sc), and BaMF_4_ compounds (M = Mg, Mn, Fe, Co, Ni, and Zn). GdMnO_3_ (GMO), a representative rare earth manganite, is a captivating material due to its intriguing properties, including a complex low-temperature magnetic arrangement, magnetic perturbation-induced ferroelectricity, and a substantial magnetoelectric coefficient [33,34].

This material exhibits meta-magnetic behavior with decreasing temperature, transitioning from a paramagnetic phase to an antiferromagnetic phase around 45 K, and then to a weak ferromagnetic (bend antiferromagnetic) phase at approximately 25 K [35,36]. Leveraging the magnetoelectric effect found in multiferroic materials, GMO holds promise for various practical applications in electrical and magnetic devices, such as memory blocks, transducers, and magnetic sensors. Unlike traditional magnetic data storage systems that rely on magnetic fields for data writing, the control of each bit’s magnetization direction can be achieved using the high leakage field of the electromagnet in the write head [36]. The utilization of the magnetoelectric effect to control magnetization using electric fields could present an innovative approach for developing high-density, low-power data storage systems. Moreover, the magnetoelectric properties offer versatile applications in sensors and transducers, as well as film waveguides, enhancing integrated optical and fiber optic communications capabilities [35,37]. Chopped strands, comprising glass fiber roving or individual glass fibers, play a vital role as reinforcements in the production of high-performance composites. Their exceptional mechanical properties make them indispensable in various industries [38]. For composites, chopped strands are a fairly affordable reinforcement that serves a variety of functions. The material is lightweight and resistant to corrosion. The characteristics of the composite rely heavily on the bond formed at the interface between the chopped strands and the matrix resin [32,39]. Chopped strands composites not only exhibit remarkable strength when compared to the other composites available, but they also hold promise as a viable material for marine applications owing to their excellent moisture resistance, electrical insulation, and fire resistance. These chopped strands are derived from continuous glass fibers and are specifically engineered to withstand the demanding bonding requirements of the fabrication process.

Thanks to their exceptional physical and chemical properties, chopped strands are widely utilized as reinforcement particles in the production of various technical textiles. Industries such as sports textiles, automotive textiles, aviation, wind turbine blades, and textile-reinforced concrete benefit from the superior attributes of chopped strands in their manufacturing processes [32,40]. In addition, the chopped strands used in polyester resin are also used in small GRP pieces for electrical applications [40]. In previous studies, the shielding effect (SE) at 12.4 GHz was measured as −29.4 dB for shielding made of, for example, a 2 mm thick CF/epoxy composite [41]. In a separate study [42], CF-PTh/graphene exhibited a shielding effect value of −30 dB in the frequency range from 12.4 to 18 GHz with a thickness of 7 × 5 mm. In another graphene-based study, it was found that the produced graphene nanocomposites showed a shielding effect value of −30 dB in the X-band [43].

In this study, for the first time, a GdMnO_3_/chopped strands composition was produced as a composite according to optimum parameters, and the electromagnetic shielding effect values of a 1.3 mm thickness are characterized in a wide frequency band. New GdMnO_3_/chopped strands composites were obtained by producing magnetic GdMnO_3_ and chopped strands by the method of mixing oxides; their components were taken in different proportions and molded with epoxy in a temperature-controlled hydraulic press. The composition of these composites was analyzed using an XRD device (Bruker D2 phaser, Germany). The microwave shielding effect values of the newly produced GdMnO_3_/chopped strands composites with a thickness of 1.3 mm were measured using a network analyzer device (N 5230A PNA Series) capable of precise measurements within the 6.5–17.5 GHz frequency range. This frequency range encompasses specific radar frequency bands such as P, L, S, and C.

## 2. Materials and Methods

### 2.1. Preparation of GdMnO_3_

GdMnO_3_ and chopped strands were combined to form a composite through a series of traditional oxide mixing processes. High-purity Gd_2_O_3_ (99.9%; Alfa Aesar, USA) and Mn_2_O_3_ (98%; Alfa Aesar, USA) powders were precisely weighed and mixed in ethyl alcohol using zirconium balls in a plastic container for 20 h at the mill. The oxide mixing process plays a key role in achieving phase homogeneity by thoroughly dispersing the reactants in ethyl alcohol via ball milling, ensuring uniform particle size and intimate contact between the oxides. This promotes effective diffusion and reaction kinetics during the heat treatment. The resulting slurry was dried in an oven at 95 °C for 24 h and then calcined at 600 °C for 10 h. The calcination at 600 °C initiated the reaction between Gd_2_O_3_ and Mn_2_O_3_, promoting phase formation while decomposing any volatile byproducts and eliminating potential intermediate phases. The calcined powders were further ground into a fine consistency and then sintered at 1350 °C for 20 h with a heating and cooling rate of 120 °C/h. Sintering at 1350 °C facilitated grain growth, densification, and crystallization, ensuring the formation of a single-phase orthorhombic GdMnO_3_ structure. Thus, the carefully optimized thermal treatment ensured high phase purity, improved crystallinity, and a defect-free microstructure, essential for enhancing the material’s electromagnetic properties. resulting in the formation of single-phase GdMnO_3_ (Figure 1). Meanwhile, the chopped strands used in the study were commercially obtained from Koloğlu Kimya, and their average nominal length ranged from 3 to 6 mm. These chopped strands were known for their high impact resistance and exceptional mechanical and machining properties. The sintered GdMnO_3_ sample and chopped strands sample were characterized using an XRD (X-Ray diffractometry-D2 Phaser Bruker AXS) instrument with Cu-Kα radiation in the 2θ range of 10–70° and scanning speed of 1°/min. X-ray powder diffractometry allowed for the identification of the main structures present, which were identified as GdMnO_3_ and chopped strands.

### 2.2. Preparation of GdMnO_3_/Chopped Strands/Epoxy Composites

The composition of GdMnO_3_: chopped strands and epoxy powder was cured and then molded into a certain shape and converted into new composites. The epoxy powder mixture of the entire composition was taken as 5/1 by weight. Molding was formed in a pressure and temperature-regulated hydraulic press device under 5 MPa pressure and 100 °C for 1 h. The incorporation of chopped strands reinforces the composite, increasing its strength, toughness, and durability, making it more resistant to wear and mechanical stress while epoxy plays a crucial role in composite formation by acting as a binding matrix that holds the GdMnO_3_ particles and chopped strands together, ensuring uniform distribution of components and providing structural cohesion. It also contributes to the flexibility and processability of the composite, allowing it to be molded under pressure and temperature while maintaining stability and durability in practical applications. In order to measure the shielding effect values, a molded 1.3 mm thick shaped composite sample was produced. To examine the microwave shielding effect properties of the newly formed GdMnO_3_/chopped strands composites at different ratios and with a thickness of 1.3 mm, a network analyzer device NA (N 5230A PNA Series) instrument was employed. The measurements were conducted in the wide frequency range of 6.5–17.5 GHz.

## 3. Results and Discussion

### 3.1. XRD Analysis of GdMnO_3_/Chopped Strands

The structural characterization of GdMnO_3_ and chopped strands was conducted using XRD spectroscopy (Bruker/Alpha-T instrument). The GdMnO_3_ compound was sintered at 1350 °C for 20 h. An X-ray diffractometry analysis of GdMnO_3_ and chopped strands detected the structure of single-phase GdMnO_3_ and the structure of chopped strands (Figure 2). As a result of XRD analysis, the main phases were determined as GdMnO_3_ (PDF card no: 00-025-0337) [7]. By applying the oxide mixing method, the single-phase structure of GdMnO_3_ was acquired by removing the possible intermediate phases with the appropriate calcination temperature and a certain sintering temperature. Based on the measurement findings, the diffraction peaks of the samples align with the structure of the main phase. Furthermore, the creation of GdMnO_3_ is significantly influenced by temperature, necessitating high temperatures for the formation of these single-phase structures.

The XRD analysis reveals that the synthesized GdMnO_3_ possesses a single-phase orthorhombic crystal structure, with no detectable secondary phases or impurities, confirming high phase purity. The diffraction peaks observed in the XRD pattern match the reference PDF card no: 00-025-0337, indicating that the desired GdMnO_3_ phase was successfully formed after sintering at 1350 °C for 20 h. The sharp and well-defined peaks further suggest a high degree of crystallinity, which is essential for optimizing the material’s electromagnetic properties. The lattice parameters of GdMnO_3_ were determined using the FullProf software, version 2.8.1, which employs Rietveld refinement to fit the experimental XRD data to a crystallographic model. This analysis provided the unit cell dimensions: a = 5.840 Å, b = 7.430 Å, c = 5.310 Å, with a unit cell volume of 230.41 Å³ in the Pnma (62) space group (Table 1) [7,44].

### 3.2. SEM Analysis of GdMnO_3_

The microstructures and morphology of orthorhombic GdMnO_3_ samples sintered at 1350 °C for 20 h were examined by scanning electron microscope (SEM, JEOL 5910LV), and phase analyses of the internal structure were performed using energy distribution spectrometry (EDS) (Figure 3). As in XRD analysis, in all samples under SEM analysis it was observed that GdMnO_3_ in orthorhombic crystal structure formed a single-phase structure and there were no secondary phases or microstructural impurities in the microstructure (Figure 3a,b). On the other hand, GdMnO_3_ is in centrosymmetric orthorhombic crystal structure.

The microstructure exhibited the formation of grains that displayed compatible morphologies with each other. The grains in the GdMnO_3_ structure have a spherical shape. EDS analysis applied to GdMnO_3_ grains gave similar results to the theoretical composition of GdMnO_3_. In accordance with the theoretical composition of GdMnO_3_ (60.44% Gd, 21.11% Mn, and 18.45% O), EDS analysis on GdMnO_3_ grains gave close results. When examined at various magnifications, it was seen that the grain size was compatible. The absence of foreign elemental peaks in the EDS spectra and the consistent grain morphology in SEM images strongly support the claim that the material achieved high phase purity (single phase), reinforcing the reliability of the synthesis method.

The electromagnetic shielding effect is influenced by the multi-reflection effect, which involves internal reflections within the material. This effect commonly occurs when there are numerous large surface areas or interface regions. Sintered GdMnO_3_ with porous structures is likely to possess a substantial specific surface area and multiple internal grain boundaries. These attributes contribute to enhancing the efficiency of the electromagnetic shielding effect.

### 3.3. EMI Shielding Measurements of GdMnO_3_/Chopped Strands

In Figure 4, the shielding effect of epoxy composites (GdMnO_3_/chopped strands) is depicted, showcasing its frequency-dependent variation across the wide 6.5–17.5 GHz frequency range. The SE values were measured using NA (N 5230A PNA Series) instrument. To ensure the 50 ohm impedance at the input and output ports of this device during measurements, coaxial holders with adequately large diameters were utilized. At first, the value of the device without a sample was measured. Then, the samples were always placed in the device in a certain order and the sample place was compressed equally at three different points in order to maintain a constant pressure during the measurement. The data for shielding effect measurements were obtained by comparing the presence and absence of samples inside the apparatus. The flanged coaxial tester used in the experiment maintained consistent diameters throughout. The holder in which the samples are placed is an elongated coaxial transmission line with special spikes and corresponding notches that maintain the typical impedance of 50 ohms along the length of the holder. Each measurement of the smooth-surfaced round 1.3 mm thick composite specimens was carefully verified and repeated. In order for the SE measurements to be as reproducible as possible, the load sample and reference sample area were of the same thickness. The observed shielding effectiveness values of the composites are influenced by their geometry and orientation. In general, the lowest shielding effectiveness value indicates that incident electromagnetic waves do not pass through the composite material, i.e., electromagnetic waves are reflected back or absorbed in the material. We look at how much of the electromagnetic wave coming to the material passes through the material. The electromagnetic shielding efficiency of the GdMnO_3_/chopped strands composite is determined by several key factors, including magnetic loss, impedance matching, multi-reflection effects, and material composition. GdMnO_3_’s strong magnetic properties contribute to high magnetic loss, which enhances electromagnetic wave absorption, particularly at higher frequencies. Impedance matching, influenced by the proportion of chopped strands, ensures effective wave penetration into the material, reducing surface reflection and increasing absorption. The multi-reflection effect, enhanced by the composite’s porous microstructure, causes incident waves to undergo multiple internal reflections, further dissipating energy and boosting attenuation. Additionally, the material composition ratio directly impacts the shielding performance, with a higher GdMnO_3_ content (80–20%) improving shielding in the higher GHz range, while a higher chopped strand content (60–40%) enhances wave scattering and reflection at lower frequencies. Among GdMnO_3_/chopped strands composites, it is understood that the microwave shielding effect performance of the epoxy- (GdMnO_3_-chopped strands (80–20% by weight)) composition is superior to other epoxy-(GdMnO_3_-chopped strands (60–40% by weight)) composites. At the specified ratio, the composite exhibited its minimum shielding effect value, reaching −32.54 dB at a frequency of 16.74 GHz (Figure 4). This composite showed −32.04, −30.63, −29.2, and 31.34 dB shielding effectiveness values at the 6.65, 9.11, 11.01, and 16.56 GHz frequencies, respectively. Furthermore, the composite as a wide band demonstrated a shielding effect value below −10 dB within the frequency range of 6.5 to 17.5 GHz. Additionally, in the frequency ranges of 0 to 6.98 GHz, 7.78 to 13.19 GHz, 13.32 to 14.55 GHz, 14.59 to 15.16 GHz, 16.07 to 16.32 GHz, 16.39 to 16.59 GHz, and 16.66 to 16.86 GHz, shielding effect values of less than −20 dB were observed. In addition, the composite exhibited a shielding effect value of less than −30 dB in the frequency range of 6.61 to 6.69 GHz.

When the amount of GdMnO_3_ decreased and the amount of chopped strands increased, that is, when the measurement tests were started for the GdMnO_3_/chopped strands (60–40% by weight) composite, the lowest shielding effect value with −35.6 dB was achieved at a frequency of about 6.9 GHz.

Figure 4 supports the claim that increasing the chopped strand content improves impedance matching and internal reflections, enhancing absorption and overall shielding effectiveness. Additionally, it highlights the frequency-dependent behavior of each composite, with the 60–40% composite maintaining an SE value below −10 dB over a wider frequency range compared to the 80–20% composite, confirming its broader and more effective shielding performance.

Additionally, this composite showed −33.71, −33.47, −29.67, and −27.06 dB shielding effect values at the 6.67, 16.43, 17.4, and 16.74 GHz frequencies, respectively. It also exhibited a wide band, displaying a shielding effect value below −10 dB within the frequency range of 6.5 to 16.93 GHz.

For the GdMnO_3_/chopped strands composite, with a weight ratio of 60–40%, in the frequency ranges of 6.5 to 6.96 GHz, 8.51 to 13.09 GHz, 13.32 to 13.84 GHz, 14.01 to 15.22 GHz, 16.05 to 16.18 GHz, and 16.74 to 16.86 GHz, shielding effect values of less than −20 dB were exhibited (Table 2). Moreover, this composite exhibited an SE value of less than −30 dB in the frequency range of 6.59–6.71 GHz.

This difference arises from the increased proportion of chopped strands, which enhances impedance matching and wave attenuation, facilitating stronger absorption of electromagnetic waves. The higher chopped strand content also increases the number of internal interfaces, promoting multiple reflections within the composite, thereby trapping and dissipating more electromagnetic energy. Conversely, the 80–20% composite, with a higher GdMnO_3_ content, exhibits a stronger magnetic loss but lower impedance matching, leading to a slightly reduced shielding performance. The optimal balance between magnetic loss, dielectric loss, and impedance matching in the 60–40% composite results in enhanced shielding efficiency, particularly in the lower GHz range [29,45].

The variations in shielding effectiveness correlate with the structural and electrical properties of the composites—GdMnO_3_’s high magnetic permeability enhances absorption at higher frequencies, while chopped strands improve impedance matching and scattering at lower frequencies. This balance of reflection and absorption dictates the frequency-dependent shielding performance of each composite.

Instead of relying on a single material to meet microwave shielding effectiveness requirements, composite materials with improved shielding effectiveness are needed. Composites with porous structures are likely to possess a large specific surface area and multiple internal grain boundaries, contributing to their electromagnetic shielding effect properties, which are characterized using the shielding effect value. The microwave shielding effect value is also influenced by the compatibility of the irradiation impedance on the material’s surface. The use of chopped strands enhances impedance matching in the transmission between components of composite materials. Consequently, sharp shielding effect peaks are observed in the measurement graph due to the resonance effect of the holder geometry and reflection. For effective electromagnetic signal shielding, the shielding material must reflect the signal back or absorb it within the material. The signal must be prevented from passing across.

Typically, the lowest SE value indicates that the electromagnetic wave passes through the material the least. That is, the incident electromagnetic wave is either reflected back or absorbed in the material. The primary purpose of electromagnetic shielding is to achieve back-reflection or the maximum absorption of incoming electromagnetic waves. This is influenced by various factors, such as the composition method, microstructure, and geometric arrangement, leading to numerous reflections and refractions of electromagnetic waves.

GdMnO_3_ enhances microwave shielding through its strong magnetoelectric properties, promoting electromagnetic wave absorption via multi-reflection and impedance matching, while its porous structure increases attenuation. Chopped strands, composed of glass fibers, reinforce the composite structurally and improve impedance matching, ensuring effective wave penetration and absorption. Their dielectric properties complement the magnetic characteristics of GdMnO_3_, optimizing both reflection and absorption mechanisms. Together, these components create a synergistic effect, allowing tunable shielding effectiveness [29,46].

The electromagnetic shielding effect material benefits from the interface polarization between the chopped strands and GdMnO_3_. The uniformity of irradiation impedance on the composite surface also plays a crucial role in determining the efficiency of the microwave shielding effect. The increase in the peak widths of the shielding effect can be associated with the observed enlargement in the grain size distribution of the GdMnO_3_ structure. The 1.3 mm thick GdMnO_3_/chopped strands composites exhibit a strong shielding effect for electromagnetic waves within the specified frequency band and at specific frequencies. Moreover, by adjusting the concentration of the components, the microwave shielding effect performance of these 1.3 mm thick GdMnO_3_/chopped fiber composites, produced for the first time in the 6.5 to 17.5 GHz frequency band, can be modified.

The weight and thickness constraints are indeed crucial factors in electromagnetic shielding materials, as they directly impact their practicality in real-world applications. In this study, the GdMnO_3_/chopped strands composites were designed with a controlled thickness of 1.3 mm, ensuring an optimal balance between shielding effectiveness and material weight. Compared to traditional shielding materials such as carbon fiber composites, which often require higher thicknesses to achieve similar shielding levels, the GdMnO_3_-based composites demonstrated a minimum shielding effectiveness of −35.61 dB at 6.9 GHz within the 6.5–17.5 GHz frequency range, making them competitive in high-frequency EMI protection [47].

The GdMnO_3_/chopped strands composites offer a scalable and cost-effective alternative to traditional EMI shielding materials, with high shielding effectiveness. The oxide mixing and hot-pressing fabrication methods can be optimized for large-scale production, reducing processing time and costs. Compared to metal-based shielding, these composites provide a balance between performance, weight, and affordability, making them suitable for aerospace, defense, consumer electronics, and automotive applications. Specifically, these materials are well-suited for high-frequency EMI protection such as military radar (X- and Ku-bands), 5G and satellite communications (C- and Ku-bands), automotive radar, and medical imaging devices. However, challenges such as mechanical durability, integration with existing products, and performance in real-world conditions must be addressed.

### 3.4. Future Research Directions and Study Limitations

Future research can enhance the microwave shielding efficiency of GdMnO_3_/chopped strands composites by focusing on material composition, frequency range optimization, and geometrical modifications. Material-wise, incorporating conductive fillers like graphene, carbon nanotubes, or MXenes could improve electrical conductivity and absorption mechanisms, leading to stronger shielding across a wider frequency range. Optimizing the GdMnO_3_-to-chopped strand ratio or introducing hybrid magnetic–dielectric materials could further fine-tune impedance matching and wave attenuation. Expanding the frequency range by modifying crystallographic structure or using dopants (e.g., Fe, Co, or rare-earth elements) may enhance performance in higher GHz bands used in advanced radar and 6G communications. Additionally, altering the geometrical arrangement, such as layered structures, graded interfaces, or porous architectures, could increase internal reflections and absorption pathways, further improving SE. These advancements could make the composite more effective for military stealth applications, high-frequency communication, and electromagnetic pollution control.

## 4. Conclusions

This study involved the fabrication of 1.3 mm thick GdMnO_3_/chopped strands composites using the oxide mixing technique, with weight ratios of 80–20% and 60–40%. Notably, this is the first investigation of shielding effect measurements within the wide 6.5–17.5 GHz frequency band for known GdMnO_3_/chopped strands composites. The microwave shielding effectiveness tests of these composites yielded highly promising results for further research, owing to their easy preparation technique and cost-effective manufacturing processes. The epoxy-(GdMnO_3_/chopped strands (60–40% by weight)) composite exhibited the lowest shielding effect value of −35.61 dB at a frequency of 6.9 GHz, with shielding effect values below −10 dB in the range of 6.5 to 16.93 GHz. It also demonstrated a shielding effect value of less than −20 dB in the frequency band from 8.51 to 13.09 GHz. On the other hand, the epoxy-(GdMnO_3_/chopped strands (80–20% by weight)) composite displayed a shielding effect value below −10 dB in the frequency region of 6.5–17.5 GHz. It also demonstrated a shielding effect value of less than −20 dB in the frequency band from 0 to 6.98 GHz and 7.78 to 13.19 GHz. This composite exhibited the lowest shielding effect value of −32.54 dB at a frequency of 16.74 GHz. These results indicate the promising potential of the GdMnO_3_/chopped strands composite for achieving desirable microwave shielding effect values. Furthermore, the comparative performance of these composites with other materials in the literature suggests the significance of their electromagnetic reflection loss and shielding effectiveness properties for various applications. In future research, exploring the microwave shielding efficiency, reflection loss, and absorption efficiency of the GdMnO_3_/chopped strands composite at other radar frequencies or higher frequency bands, using different polymers in varying proportions and geometrical arrangements, could offer valuable insights. Such composites may serve as valuable resources for preventing electromagnetic pollution and developing armor and electromagnetic shielding effect materials.

## Figures and Tables

**Figure 1 materials-18-00863-f001:**
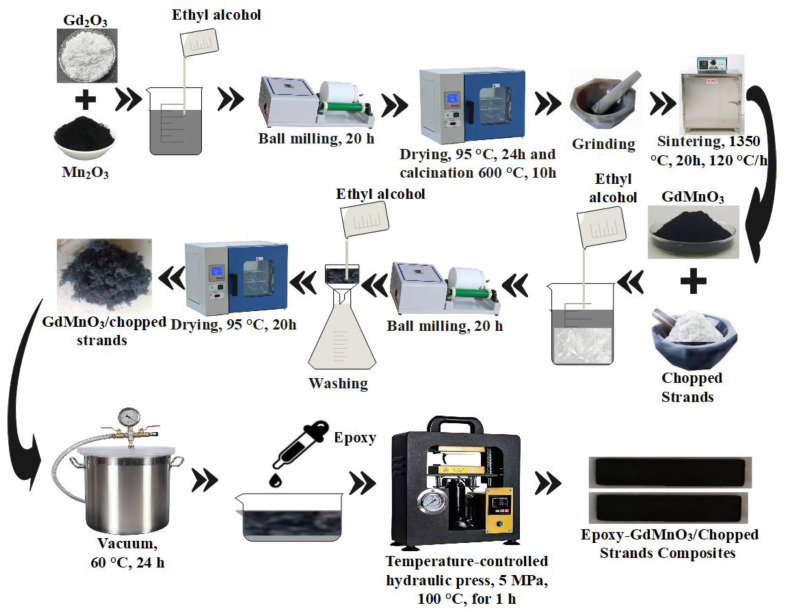
Flow diagram of the synthesis process.

**Figure 2 materials-18-00863-f002:**
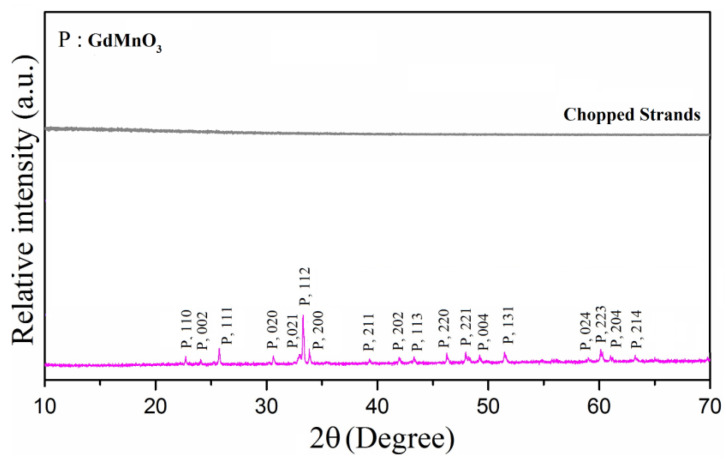
X-ray diffractogram of single-phase GdMnO_3_ heat-treated for 20 h at 1350 °C.

**Figure 3 materials-18-00863-f003:**
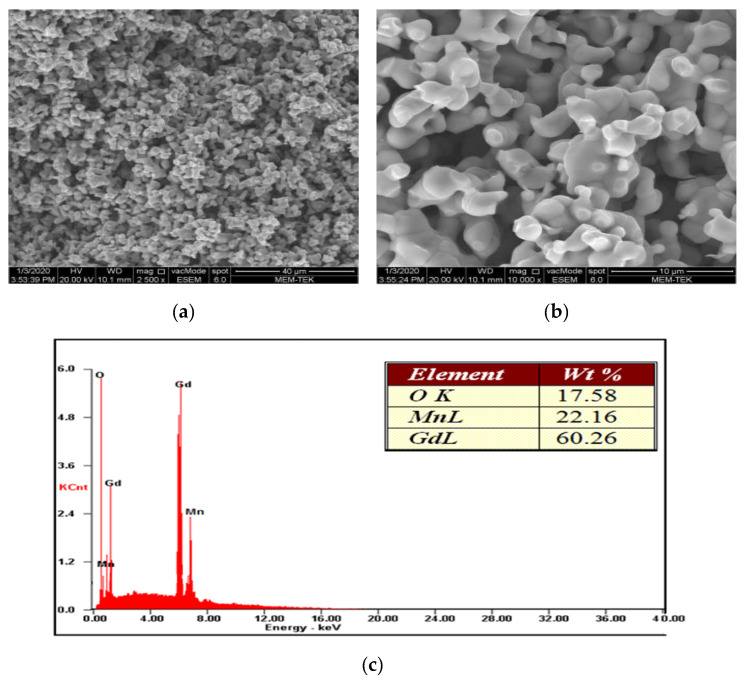
SEM photographs of single-phase GdMnO_3_ heat-treated for 20 h at 1350 °C (**a**) at ×2500, (**b**) at ×10,000, and (**c**) EDS examination of GdMnO_3_ at ×10,000.

**Figure 4 materials-18-00863-f004:**
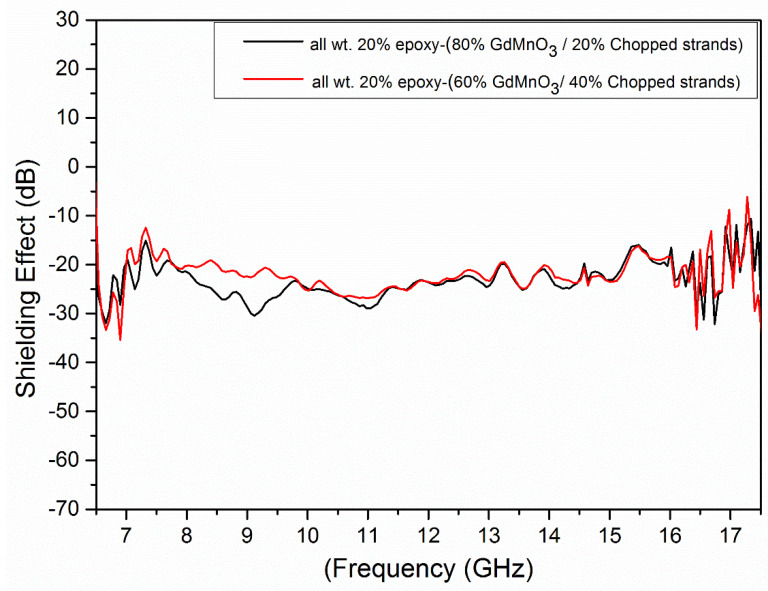
Shielding effect of the epoxy-(GdMnO_3_/chopped strands) composites: All wt. 20% epoxy- (80% GdMnO_3_/20% chopped strands) compositions, all wt. 20% epoxy-(60% GdMnO_3_/40% chopped strands) compositions.

**Table 1 materials-18-00863-t001:** Using FullProf, computed lattice parameters of a GdMnO_3_ specimen.

Sample	a (Å)	b (Å)	c (Å)	Volume (Å3)	Space Group
GdMnO_3_	5.840	7.430	5.310	230.41	Pnma (62)

**Table 2 materials-18-00863-t002:** The produced samples and their shielding effect values at different frequencies.

Sample	SE (dB)	Frequency (GHz)
All wt. 20% epoxy- (80% GdMnO_3_/20% chopped strands)	−32.54	16.74
	−32.04	6.65
	−30.63	9.11
	−29.2	11.01
	−31.34	16.56
	−10	6.5–17.5
	−20	0–6.98
		7.78–13.19
		13.32–14.55
		14.59–15.16
		16.07–16.32
		16.39–16.59
		16.66–16.86
	−30	6.61–6.69
All wt. 20% epoxy- (60% GdMnO_3_/40% chopped strands)	−35.61	6.9
	−33.71	6.67
	−33.47	16.43
	−29.67	17.4
	−27.06	16.74
	−10	6.5–16.93
	−20	6.5–6.96
		8.51–13.09
		13.32–13.84
		14.01–15.22
		16.05–16.18
		16.74–16.86
	−30	6.59–6.71

## Data Availability

The original contributions presented in this study are included in the article. Further inquiries can be directed to the corresponding author.

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
