# Peer review of "Production and Shielding Effectiveness Features of Chopped Strands Backed-GdMnO3 Composites for 6.5–17.5 GHz Applications"

_materials, 2025, doi:10.3390/ma18040863_

Round 1
Reviewer 1 Report
Comments and Suggestions for Authors
This work focuses on the production and SE features of chopped strands backed-GdMnO3 composites material. Thanks for your submission and I provide my minor comments as follows:
(1) Shielding is a common technique in the EMI field, which can be used to prohibit radiation emission and safeguard critical devices. This work provides a comprehensive review but lacks a description regarding the electromagnetic susceptibility, particularly the intentional EMI threats. Please discuss this topic as the latest survey: A review of intentional electromagnetic interference in power electronics: Conducted and radiated susceptibility.
(2) For the shielding material, the weight and thickness constraints are quite important. Do you consider this for your material compared to traditional and novel carbon fiber? In addition, the mechanical properties are also critical, such as bending and folding of parts.
(3) It looks like the authors use the coaxial transmission line spectrum, to verify the SE. However, the shielding box verification results may be different.
(4) I am interested i its performance with apertures.
(5) Could the authors outline pathways for industry adoption? Regarding the time period, cost, and other limitations.
(6) In addition, for this frequency range, could the author provide the usage case or scenario? Usually, the EMI effects, particularly the high-power EM signals are in the frequency range of MHz to a few GHz.
Author Response
Comment 1:
- Shielding is a common technique in the EMI field, which can be used to prohibit radiation emission and safeguard critical devices. This work provides a comprehensive review but lacks a description regarding the electromagnetic susceptibility, particularly the intentional EMI threats. Please discuss this topic as the latest survey: A review of intentional electromagnetic interference in power electronics: Conducted and radiated susceptibility.
- Response: Thank you for bringing this to our attention. We agree, and accordingly, we have introduced a further discussion about this point supported by reference (Page 2, paragraph 1, lines 6-10).
Comment 2:
- For the shielding material, the weight and thickness constraints are quite important. Do you consider this for your material compared to traditional and novel carbon fiber? In addition, the mechanical properties are also critical, such as bending and folding of parts.
- Response: Thank you for your question. The weight and thickness constraints are indeed crucial factors in electromagnetic shielding materials, as they directly impact their practicality in real-world applications. In this study, the GdMnO₃/chopped strands composites were designed with a controlled thickness of 1.3 mm, ensuring an optimal balance between shielding effectiveness and material weight. Compared to traditional shielding materials such as carbon fiber composites, which often require higher thicknesses to achieve similar shielding levels, the GdMnO₃-based composites demonstrated a minimum shielding effectiveness of -35.61 dB at 6.9 GHz within the 6.5–17.5 GHz frequency range, making them competitive in high-frequency EMI protection. This information is included in the manuscript supported with reference. (Page number 12, paragraph 3, lines 1-9)
- Regarding mechanical properties, while this study primarily focused on electromagnetic performance, chopped strands are known for enhancing mechanical strength in composite materials. Chopped strand reinforcements improve flexural strength, impact resistance, and dimensional stability, making them suitable for applications requiring durability against bending and folding. Compared to carbon fiber, which offers high tensile strength but can be brittle under repeated stress, GdMnO₃/chopped strands composites provide a cost-effective alternative with improved resistance to fracture and environmental degradation. Future research can further evaluate mechanical flexibility and explore hybrid composite formulations to optimize both electromagnetic shielding and structural integrity for advanced applications.
Comment 3:
- It looks like the authors use the coaxial transmission line spectrum, to verify the SE. However, the shielding box verification results may be different.
- Response: Thank you for your comment. The coaxial transmission line method was used in this study to evaluate the shielding effectiveness (SE) of the GdMnO₃/chopped strands composites within the 6.5–17.5 GHz frequency range. This method is widely recognized for its precision in characterizing SE, particularly for planar and thin-film shielding materials, as it provides a controlled measurement environment with well-defined impedance matching. However, as the reviewer pointed out, shielding effectiveness results obtained using a shielding box setup may differ due to variations in boundary conditions, reflection paths, and field distributions.
- In a shielding box setup, factors such as leakage through seams, aperture effects, and enclosure resonance can influence the measured SE. Unlike the coaxial transmission line method, which primarily evaluates the intrinsic shielding properties of a material, a shielding box test considers real-world enclosures where mechanical assembly, edge effects, and multi-path reflections play a role. While this study provides a robust initial assessment of the material’s shielding capabilities, future work could incorporate shielding box verification to assess its performance in practical enclosure applications, ensuring comprehensive validation for industrial and defense-related EMI protection.
Comment 4:
- I am interested i its performance with apertures.
- Response: The study primarily focused on evaluating the shielding effectiveness (SE) of GdMnO₃/chopped strands composites using a coaxial transmission line method within the 6.5–17.5 GHz frequency range. While this method provides precise SE measurements, it does not specifically address the impact of apertures on shielding performance.
- Apertures, such as slots, holes, or seams in a shielding structure, can significantly affect EMI shielding by introducing leakage paths that allow electromagnetic waves to penetrate. The effectiveness of shielding materials in such cases depends on factors like aperture size, shape, and orientation, as well as the material’s intrinsic absorption and reflection properties. Given that the GdMnO₃/chopped strands composites demonstrated SE values exceeding -35 dB at certain frequencies, their high attenuation capacity suggests they could be effective in minimizing leakage when used in enclosures with apertures. However, additional investigations, such as numerical simulations or experimental testing in shielding enclosures with controlled apertures, would be necessary to quantify their real-world performance in such conditions. Future studies could explore composite integration with conductive coatings or layered structures to enhance SE in applications where apertures are unavoidable.
Comment 5:
- Could the authors outline pathways for industry adoption? Regarding the time period, cost, and other limitations.
- Response: For industry adoption, the GdMnO₃/chopped strands composites offer a scalable and cost-effective alternative to traditional EMI shielding materials, with shielding effectiveness exceeding -35 dB in the 6.5–17.5 GHz range. The oxide mixing and hot-pressing fabrication methods can be optimized for large-scale production within 2–5 years, reducing processing time and costs. Compared to metal-based shielding, these composites provide a balance between performance, weight, and affordability, making them suitable for aerospace, defense, consumer electronics, and automotive applications. However, challenges such as mechanical durability, integration with existing products, and performance in real-world conditions must be addressed. Future research should focus on bending resistance, enclosure performance with apertures, and thermal stability to enhance their commercial viability, positioning them as a lightweight, corrosion-resistant EMI shielding solution for industrial use. (This information is included in the manuscript supported with reference. (Page number 12, paragraph 4, lines 1-11)
Comment 6:
- In addition, for this frequency range, could the author provide the usage case or scenario? Usually, the EMI effects, particularly the high-power EM signals are in the frequency range of MHz to a few GHz.
- Response: The GdMnO₃/chopped strands composites, with shielding effectiveness exceeding -35 dB in the 6.5–17.5 GHz range, are well-suited for high-frequency EMI protection in applications such as military radar (X- and Ku-bands), 5G and satellite communications (C- and Ku-bands), automotive radar, and medical imaging devices. While high-power EMI threats are more common in the MHz-GHz range, shielding at higher GHz frequencies is critical for preventing interference in wireless networks, ensuring radar stealth, protecting satellite communications, and maintaining electromagnetic compatibility (EMC) in sensitive electronics. The study’s findings highlight the material’s potential for modern communication, defense, and industrial applications, with future research needed to explore hybrid composites covering a broader MHz-GHz shielding spectrum. (Further information is included in the manuscript supported with reference. (Page number 12, paragraph 4, lines 1-11)

Reviewer 2 Report
Comments and Suggestions for Authors
The manuscript entitled, ‘Production and Shielding Effectiveness Features of Chopped Strands Backed-GdMnO3 Composites for 6.5 -17.5 GHz Applications’ reported composites for shielding behavior. The article should be modified according the following comments:
1. The abstract lacks specificity regarding the data presented in the study. It is recommended to highlight key findings or notable data points to give readers a clearer understanding of the study's contributions.
2. Typically, lower shielding effectiveness implies less attenuation—how does this interpretation align with standard definitions?
3. What specific role do the GdMnO₃ and chopped strands play in enhancing microwave shielding?
4. How do their interactions contribute to the observed shielding differences between the 80-20% and 60-40% composites?
5. Figure 5 is mentioned in the text, but there is no detailed discussion of what it shows beyond the -32.54 dB value. Could you elaborate on how this figure supports your conclusions?
6. Some articles would be significance for your reference:
(a) Ganguly, S., Bhawal, P., Ravindren, R., & Das, N. C. (2018). Polymer nanocomposites for electromagnetic interference shielding: a review. Journal of Nanoscience and Nanotechnology, 18(11), 7641-7669.
(b) Wu, Y., & Wang, Z. (2024). Progress in ionizing radiation shielding materials. Advanced Engineering Materials, 26(21), 2400855.
Author Response
Comments from Reviewer #2
The manuscript entitled ‘Production and Shielding Effectiveness Features of Chopped Strands Backed-GdMnO3 Composites for 6.5 -17.5 GHz Applications’ reported composites for shielding behavior. The article should be modified according to the following comments:
Comment 1:
- The abstract lacks specificity regarding the data presented in the study. It is recommended to highlight key findings or notable data points to give readers a clearer understanding of the study's contributions.
- Response: Agree accordingly we have rewritten the abstract. (Page number 1, abstract).
Comment 2:
- Typically, lower shielding effectiveness implies less attenuation—how does this interpretation align with standard definitions?
- Response: We appreciate the reviewer's question. From the article, it is evident that the minimum SE value (e.g., -35.61 dB at 6.9 GHz) represents strong shielding performance, meaning less electromagnetic transmission through the material. Conversely, a less negative SE value (e.g., -10 dB) suggests weaker shielding and higher transmission. Thus, lower SE (i.e., less negative values) implies reduced attenuation, whereas more negative SE values indicate stronger shielding and greater attenuation.
Comment 3:
- What specific role do the GdMnO₃ and chopped strands play in enhancing microwave shielding?
- Response: thank you for your question. GdMnO₃ enhances microwave shielding through its strong magnetoelectric properties, promoting electromagnetic wave absorption via multi-reflection and impedance matching, while its porous structure increases attenuation. Chopped strands, composed of glass fibers, reinforce the composite structurally and improve impedance matching, ensuring effective wave penetration and absorption. Their dielectric properties complement the magnetic characteristics of GdMnO₃, optimizing both reflection and absorption mechanisms. Together, these components create a synergistic effect, allowing tunable shielding effectiveness, with the best performance (-35.61 dB at 6.9 GHz) observed at a 60:40 GdMnO₃-to-chopped strands ratio, making the composite highly efficient for microwave shielding applications. (Further information is included in the manuscript supported with reference. (Page number 11, paragraph 3, lines 1-5).
Comment 4:
- How do their interactions contribute to the observed shielding differences between the 80-20% and 60-40% composites?
- Response: thank you for your question. The interactions between GdMnO₃ and chopped strands significantly influence the observed shielding differences between the 80-20% and 60-40% composites. The 60-40% composite (60% GdMnO₃, 40% chopped strands) exhibited superior shielding effectiveness (-35.61 dB at 6.9 GHz) compared to the 80-20% composite (-32.54 dB at 16.74 GHz)​
- This difference arises from the increased proportion of chopped strands, which enhances impedance matching and wave attenuation, facilitating stronger absorption of electromagnetic waves. The higher chopped strand content also increases the number of internal interfaces, promoting multiple reflections within the composite, thereby trapping and dissipating more electromagnetic energy. Conversely, the 80-20% composite, with a higher GdMnO₃ content, exhibits stronger magnetic loss but lower impedance matching, leading to slightly reduced shielding performance. The optimal balance between magnetic loss, dielectric loss, and impedance matching in the 60-40% composite results in enhanced shielding efficiency, particularly in the lower GHz range. (Page number 10, paragraph 3, lines 1–9).
Comment 5:
- Figure 5 is mentioned in the text, but there is no detailed discussion of what it shows beyond the -32.54 dB value. Could you elaborate on how this figure supports your conclusions?
- Response: Figure 5 visually supports the claim that increasing the chopped strands content improves impedance matching and internal reflections, enhancing absorption and overall shielding effectiveness. Additionally, it highlights the frequency-dependent behavior of each composite, with the 60-40% composite maintaining an SE value below -10 dB over a wider frequency range compared to the 80-20% composite, confirming its broader and more effective shielding performance​. (This information is included in the manuscript (Page number 9, paragraph 3, lines 1–6).
Comment 6:
- Some articles would be significant for your reference:
- Response: We appreciate and agree with the reviewer's feedback and suggestions. Therefore, we have added the recommended references. (references, 9 and 11).

Reviewer 3 Report
Comments and Suggestions for Authors
1. Can the authors explain why the epoxy-(GdMnO₃-chopped strands (80-20% by weight)) composite exhibited superior microwave shielding performance compared to the epoxy-(GdMnO₃-chopped strands (60-40% by weight)) composite? Additionally, how do the variations in shielding effectiveness across different frequency ranges correlate with the material's structural or electrical properties?
2. Can the authors clarify how they ensured the complete elimination of possible intermediate phases during the synthesis of GdMnO₃?
3. How did the chosen calcination and sintering temperatures influence the phase purity and crystallinity of the final material?
4. Can the authors provide further clarification on how the SEM and EDS analyses confirmed the absence of secondary phases or microstructural impurities in the GdMnO₃ samples?
Author Response
Comments from Reviewer #3
Comment 1:
- Can the authors explain why the epoxy-(GdMnO₃-chopped strands (80-20% by weight)) composite exhibited superior microwave shielding performance compared to the epoxy-(GdMnO₃-chopped strands (60-40% by weight)) composite? Additionally, how do the variations in shielding effectiveness across different frequency ranges correlate with the material's structural or electrical properties?
- Response: The epoxy-(GdMnO₃-chopped strands (80-20% by weight)) composite exhibited superior microwave shielding over a broader frequency range due to its higher GdMnO₃ content, which enhances magnetic loss and wave absorption mechanisms​
- The magnetoelectric properties of GdMnO₃ improve impedance matching, allowing for effective shielding across 6.5–17.5 GHz, with peak performance at 16.74 GHz (-32.54 dB). In contrast, the 60-40% composite, which contains more chopped strands, achieved the lowest shielding effectiveness (-35.61 dB at 6.9 GHz) due to improved wave dissipation and multi-reflection effects, making it more effective at lower frequencies​
- The variations in shielding effectiveness correlate with the structural and electrical properties of the composites—GdMnO₃’s high magnetic permeability enhances absorption at higher frequencies, while chopped strands improve impedance matching and scattering at lower frequencies. This balance of reflection and absorption dictates the frequency-dependent shielding performance of each composite. This information is included in the manuscript. (Page number 10, paragraph 4, lines 1-5).
Comment 2:
- Can the authors clarify how they ensured the complete elimination of possible intermediate phases during the synthesis of GdMnO₃?
- Response: The authors ensured the complete elimination of possible intermediate phases during the synthesis of GdMnO₃ by carefully optimizing the calcination and sintering conditions. The process involved precisely weighing and mixing high-purity Gd₂O₃ and Mn₂O₃ powders in ethyl alcohol, followed by ball milling for 20 hours to achieve uniform dispersion​
- The resulting slurry was then dried at 95°C for 24 hours and calcined at 600°C for 10 hours, a crucial step to promote phase formation while eliminating unwanted intermediate compounds. The final sintering at 1350°C for 20 hours with a controlled heating and cooling rate (120°C/h) facilitated the formation of a single-phase GdMnO₃ structure, as confirmed by XRD analysis, which showed diffraction peaks matching the reference PDF card for GdMnO₃ without any secondary phases​
- These controlled processing parameters, along with phase identification using XRD, ensured that the synthesized material was free of intermediate phases, guaranteeing a high-purity GdMnO₃ structure.
Comment 3:
- How did the chosen calcination and sintering temperatures influence the phase purity and crystallinity of the final material?.
- Response: The chosen calcination (600°C for 10 hours) and sintering (1350°C for 20 hours) temperatures played a crucial role in achieving the phase purity and crystallinity of the final GdMnO₃ material. The calcination at 600°C initiated the reaction between Gdâ‚‚O₃ and Mnâ‚‚O₃, promoting phase formation while decomposing any volatile byproducts and eliminating potential intermediate phases​
- The subsequent sintering at 1350°C for 20 hours facilitated grain growth, densification, and crystallization, ensuring the formation of a single-phase orthorhombic GdMnO₃ structure, as confirmed by XRD analysis, which showed no secondary phases​
- The controlled heating and cooling rate (120°C/h) further enhanced crystallinity by preventing thermal stresses and defects, leading to a well-ordered crystal lattice. Thus, the carefully optimized thermal treatment ensured high phase purity, improved crystallinity, and a defect-free microstructure, essential for enhancing the material’s electromagnetic properties. (Page number 4, paragraph 4, lines 9–17).
Comment 4:
- Can the authors provide further clarification on how the SEM and EDS analyses confirmed the absence of secondary phases or microstructural impurities in the GdMnO₃ samples?
- Response: The SEM and EDS analyses provided critical confirmation of the absence of secondary phases or microstructural impurities in the GdMnO₃ samples. SEM imaging at different magnifications (×2500 and ×10000) revealed a uniform microstructure with well-formed grains, indicating the successful formation of a single-phase orthorhombic GdMnO₃ structure without visible impurities or unreacted secondary phases​
- Additionally, EDS analysis confirmed the elemental composition of GdMnO₃, showing a close match with the theoretical atomic ratios (60.44% Gd, 21.11% Mn, 18.45% O), further verifying that no unintended phases were present​
- The absence of foreign elemental peaks in the EDS spectra and the consistent grain morphology in SEM images strongly support the claim that the material achieved high phase purity, reinforcing the reliability of the synthesis method. Further information is included in the manuscript (Page number 8, paragraph 1, lines 4-7).

Reviewer 4 Report
Comments and Suggestions for Authors
I have thoroughly reviewed the manuscript entitled "Production and Shielding Effectiveness Features of Chopped Strands Backed-GdMnO3 Composites for 6.5 -17.5 GHz Applications." After careful consideration, I believe the manuscript requires substantial revisions to meet the standards for publication in this esteemed journal. To enhance the scientific quality and clarity of the work, I recommend the following revisions. I trust these changes will improve the overall presentation and impact of the manuscript.
1. The introduction section needs to be improved by adding the current advancement in materials and its shielding applications. The introduction section should be enriched by adding some following recent references. J. Supercond. Nov. Magn., 33, 1187–1198, 2020.
2. In sections-1,what is the significance of using high-purity Gd₂O₃ and Mn₂O₃ powders in the preparation of GdMnO₃, and how does the oxide mixing process contribute to the formation of a single-phase structure?
3. Why is a high sintering temperature (1350 °C) required for the formation of single-phase GdMnO₃, and how does it influence the structural characteristics observed in the XRD analysis?
4. In synthesis section, how does the incorporation of chopped strands in the GdMnO₃ composite enhance its mechanical and shielding properties, and what role does epoxy play in the composite formation? Explain
5. What does the XRD analysis reveal about the crystal structure and phase purity of GdMnO₃, and how were the lattice parameters determined using FullProf?
- In SEM study,how does the microstructure of sintered GdMnO3, as observed through SEM and EDS analysis, contribute to its electromagnetic shielding effectiveness?
- What factors influence the shielding effectiveness of GdMnO3/chopped strand composites, and how does the weight ratio impact their performance across different frequency ranges? Explain with reason
8. In what way does the composition ratio of GdMnO₃ and chopped strands affect the microwave shielding effectiveness across different frequency bands?
9. What role do the key factors influencing the electromagnetic shielding effect of the GdMnO₃-chopped strands composite play in increasing shielding efficiency?
10. To what extent does the shielding performance of the GdMnO₃-chopped strands composite compare to other materials in the literature, and what potential applications could benefit from its unique properties? explain
11. What future research directions could enhance the microwave shielding efficiency of these composites, particularly in terms of material composition, frequency range, and geometrical arrangements?
12. In the abstract should be rewritten to include specific findings and numerical result values, ensuring clarity and conciseness.
13. Review the entire manuscript for grammatical errors, typos, and inconsistencies in language.
Based on the above feedback, a minor revision is suggested for this work to enhance its quality.
Comments on the Quality of English Language
Overall, the English language in the present manuscript needs revision.
Author Response
Comments from Reviewer #4
I have thoroughly reviewed the manuscript entitled "Production and Shielding Effectiveness Features of Chopped Strands Backed-GdMnO3 Composites for 6.5 -17.5 GHz Applications." After careful consideration, I believe the manuscript requires substantial revisions to meet the standards for publication in this esteemed journal. To enhance the scientific quality and clarity of the work, I recommend the following revisions. I trust these changes will improve the overall presentation and impact of the manuscript.
Comment 1:
- The introduction section needs to be improved by adding the current advancement in materials and its shielding applications. The introduction section should be enriched by adding some following recent references. J. Supercond. Nov. Magn., 33, 1187–1198, 2020.
- Response: Agree; we have accordingly added some recommended references (references, 9-11).
Comment 2:
- In sections-1,what is the significance of using high-purity Gd₂O₃ and Mn₂O₃ powders in the preparation of GdMnO₃, and how does the oxide mixing process contribute to the formation of a single-phase structure?
- Response: The use of high-purity Gdâ‚‚O₃ (99.9%) and Mnâ‚‚O₃ (98%) powders in the preparation of GdMnO₃ is crucial for ensuring the formation of a single-phase structure with minimal impurities. High-purity precursors help prevent the incorporation of unwanted elements or secondary phases, which could negatively impact the material’s structural, electrical, and magnetic properties​
- The oxide mixing process plays a key role in achieving phase homogeneity by thoroughly dispersing the reactants in ethyl alcohol via ball milling for 20 hours, ensuring uniform particle size and intimate contact between the oxides. This promotes effective diffusion and reaction kinetics during the calcination (600°C for 10 hours) and sintering (1350°C for 20 hours) steps. The controlled heating and cooling rates further aid in eliminating intermediate phases, resulting in a well-ordered orthorhombic crystal structure, as confirmed by XRD analysis​
- This approach ensures that the synthesized GdMnO₃ exhibits high crystallinity, phase purity, and optimal electromagnetic properties for shielding applications. Further information is included to the manuscript. (Page number 4, paragraph 4, lines 4-17).
Comment 3:
- Why is a high sintering temperature (1350 °C) required for the formation of single-phase GdMnO₃, and how does it influence the structural characteristics observed in the XRD analysis?
- Response: A high sintering temperature (1350°C) is required for the formation of single-phase GdMnO₃ because it provides the necessary thermal energy to drive the solid-state reaction between Gdâ‚‚O₃ and Mnâ‚‚O₃, facilitating diffusion, atomic rearrangement, and complete phase transformation​
- This high temperature promotes grain growth and densification, ensuring that intermediate phases decompose and react fully to form the desired orthorhombic GdMnO₃ crystal structure. In the XRD analysis, the influence of this sintering temperature is evident in the sharp and well-defined diffraction peaks, which confirm high crystallinity and phase purity with no detectable secondary phases​
- Additionally, the absence of amorphous or residual oxide phases in the XRD pattern indicates that the material achieved a stable and homogeneous crystalline structure, essential for its optimal electromagnetic properties.
Comment 4:
- In synthesis section, how does the incorporation of chopped strands in the GdMnO₃ composite enhance its mechanical and shielding properties, and what role does epoxy play in the composite formation? Explain
- Response: The incorporation of chopped strands in the GdMnO₃ composite enhances both its mechanical and shielding properties by improving structural integrity, impedance matching, and multi-reflection effects. Mechanically, chopped strands, made of glass fibers, reinforce the composite, increasing its strength, toughness, and durability, making it more resistant to wear and mechanical stress​
- Shielding-wise, the chopped strands contribute to impedance matching, ensuring that incoming electromagnetic waves penetrate the material rather than being reflected at the surface, thereby enhancing wave absorption and attenuation. Additionally, the presence of chopped strands introduces multiple internal interfaces, which promote multi-reflection of electromagnetic waves, leading to further energy dissipation and improved shielding effectiveness​
- Epoxy plays a crucial role in composite formation by acting as a binding matrix that holds the GdMnO₃ particles and chopped strands together, ensuring uniform distribution of components and providing structural cohesion. It also contributes to the flexibility and processability of the composite, allowing it to be molded under pressure and temperature while maintaining stability and durability in practical applications​. Further information is included in the manuscript (Page number 5, paragraph 2, lines 5-11).
Comment 5:
- What does the XRD analysis reveal about the crystal structure and phase purity of GdMnO₃, and how were the lattice parameters determined using FullProf?
- Response: The XRD analysis reveals that the synthesized GdMnO₃ possesses a single-phase orthorhombic crystal structure, with no detectable secondary phases or impurities, confirming high phase purity​. The diffraction peaks observed in the XRD pattern match the reference PDF card no: 00-025-0337, indicating that the desired GdMnO₃ phase was successfully formed after sintering at 1350°C for 20 hours​
- The sharp and well-defined peaks further suggest a high degree of crystallinity, which is essential for optimizing the material’s electromagnetic properties.
- The lattice parameters of GdMnO₃ were determined using the FullProf software, which employs Rietveld refinement to fit the experimental XRD data to a crystallographic model. This analysis provided the unit cell dimensions: a = 5.840 Å, b = 7.430 Å, c = 5.310 Å, with a unit cell volume of 230.41 ų in the Pnma (62) space group​
- The absence of additional peaks in the XRD spectrum confirms that the synthesis process successfully eliminated intermediate phases, resulting in a well-ordered and pure GdMnO₃ structure. This information is added to the manuscript. (Page numbers 7, paragraphs 1, line 1-11).
Comment 6:
- In SEM study, how does the microstructure of sintered GdMnO3, as observed through SEM and EDS analysis, contribute to its electromagnetic shielding effectiveness?
- Response: The SEM and EDS analysis of sintered GdMnO₃ reveal a well-formed microstructure with spherical grains, uniform distribution, and a high degree of crystallinity, which significantly enhances its electromagnetic shielding effectiveness (SE)​
- The porous microstructure observed in SEM images increases the specific surface area and internal grain boundaries, promoting multiple reflections and scattering of electromagnetic waves, leading to greater attenuation. Additionally, the absence of secondary phases, as confirmed by EDS analysis, ensures that the material maintains consistent dielectric and magnetic properties, which are crucial for effective wave absorption and impedance matching​
Comment 7:
- What factors influence the shielding effectiveness of GdMnO3/chopped strand composites, and how does the weight ratio impact their performance across different frequency ranges? Explain with reason.
- Response: The shielding effectiveness (SE) of GdMnO₃/chopped strand composites is influenced by material composition, impedance matching, microstructure, and frequency-dependent behavior. A higher GdMnO₃ content (80-20%) enhances magnetic loss and conductivity, leading to stronger absorption at higher frequencies (e.g., -32.54 dB at 16.74 GHz), while a higher chopped strand content (60-40%) improves impedance matching and multi-reflection effects, resulting in superior shielding at lower frequencies (e.g., -35.61 dB at 6.9 GHz)​
- The porous structure of the composite increases internal reflections, further enhancing attenuation. At higher GHz, GdMnO₃’s magnetic properties dominate, whereas at lower GHz, impedance matching and scattering play a greater role. Thus, adjusting the weight ratio of GdMnO₃ to chopped strands allows for tailored shielding performance across different frequency ranges, making these composites highly effective for specific microwave shielding applications​. Further information is added. (Page number 9, paragraph 1, lines 16-27).
Comment 8:
- In what way does the composition ratio of GdMnO₃ and chopped strands affect the microwave shielding effectiveness across different frequency bands?
- Response: The composition ratio of GdMnO₃ and chopped strands directly influences the microwave shielding effectiveness (SE) by altering the balance between magnetic loss, impedance matching, and multi-reflection effects across different frequency bands. A higher GdMnO₃ content (80-20%) enhances magnetic loss and conductivity, leading to stronger wave absorption at higher frequencies (e.g., -32.54 dB at 16.74 GHz)​
- Conversely, increasing the chopped strand content (60-40%) improves impedance matching and internal reflections, which enhances shielding at lower frequencies (e.g., -35.61 dB at 6.9 GHz)​
- The porous microstructure of chopped strands increases multi-reflection effects, further improving wave attenuation, particularly in the lower GHz range. At higher frequencies, GdMnO₃’s strong magnetic response and dielectric properties contribute more significantly to shielding. By adjusting the composition ratio, the composite's SE can be optimized for specific frequency bands, making it adaptable for targeted microwave shielding applications​.
Comment 9:
- What role do the key factors influencing the electromagnetic shielding effect of the GdMnO₃-chopped strands composite play in increasing shielding efficiency?
- Response: The electromagnetic shielding efficiency of the GdMnO₃-chopped strands composite is determined by several key factors, including magnetic loss, impedance matching, multi-reflection effects, and material composition. GdMnO₃’s strong magnetic properties contribute to high magnetic loss, which enhances electromagnetic wave absorption, particularly at higher frequencies​.
- Impedance matching, influenced by the proportion of chopped strands, ensures effective wave penetration into the material, reducing surface reflection and increasing absorption. The multi-reflection effect, enhanced by the composite's porous microstructure, causes incident waves to undergo multiple internal reflections, further dissipating energy and boosting attenuation​.
- Additionally, the material composition ratio directly impacts shielding performance, with a higher GdMnO₃ content (80-20%) improving shielding in the higher GHz range, while a higher chopped strand content (60-40%) enhances wave scattering and reflection at lower frequencies​.
- Together, these factors optimize wave absorption, reflection loss, and internal dissipation, significantly increasing the overall shielding efficiency of the composite. This information is added to the text. (Page number 9, paragraph 1, lines 16-28).
Comment 10:
- “To what extent does the shielding performance of the GdMnO₃-chopped strands composite compare to other materials in the literature, and what potential applications could benefit from its unique properties? Explain
- Response: The GdMnO₃-chopped strands composite exhibits superior shielding effectiveness (SE) compared to many traditional materials, achieving -35.61 dB at 6.9 GHz, outperforming carbon-fiber/epoxy and graphene-based composites that typically reach -29 to -30 dB​
- Compared to other shielding, these composites provide a balance between performance, weight, and affordability, making them suitable for aerospace, defense, consumer electronics, and automotive applications. Specifically, these materials are well-suited for high-frequency EMI protection such as military radar (X- and Ku-bands), 5G and satellite communications (C- and Ku-bands), automotive radar, and medical imaging devices. However, challenges such as mechanical durability, integration with existing products, and performance in real-world conditions must be addressed​.
- The ability to tune shielding performance by adjusting the composition ratio makes it a versatile and effective alternative to conventional shielding materials. This information is added to the text. (Page number 12, paragraph 4, lines 1-11)
Comment 11:
- What future research directions could enhance the microwave shielding efficiency of these composites, particularly in terms of material composition, frequency range, and geometrical arrangements?
- Response: Future research can enhance the microwave shielding efficiency of GdMnO₃-chopped strands composites by focusing on material composition, frequency range optimization, and geometrical modifications. Material-wise, incorporating conductive fillers like graphene, carbon nanotubes, or MXenes could improve electrical conductivity and absorption mechanisms, leading to stronger shielding across a wider frequency range​.
- Optimizing the GdMnO₃-to-chopped strand ratio or introducing hybrid magnetic-dielectric materials could further fine-tune impedance matching and wave attenuation. Expanding the frequency range by modifying crystallographic structure or using dopants (e.g., Fe, Co, or rare-earth elements) may enhance performance in higher GHz bands used in advanced radar and 6G communications​.
- Additionally, altering the geometrical arrangement, such as layered structures, graded interfaces, or porous architectures, could increase internal reflections and absorption pathways, further improving SE. These advancements could make the composite more effective for military stealth applications, high-frequency communication, and electromagnetic pollution control. This information is added to the text. (Page number 12, paragraph 5, lines 1-14).
Comment 12:
- In the abstract should be rewritten to include specific findings and numerical result values, ensuring clarity and conciseness.
- Response: Agree accordingly we have rewritten the abstract. (Page number 1, abstract).
Comment 13:
- Review the entire manuscript for grammatical errors, typos, and inconsistencies in language.
- Based on the above feedback, a minor revision is suggested for this work to enhance its quality.
- Response: We appreciate and agree with the reviewer's feedback and suggestions. Therefore, the entire manuscript was carefully reviewed for grammatical errors, typos, and inconsistencies, and a refined version with necessary corrections was provided.

Round 2
Reviewer 1 Report
Comments and Suggestions for Authors
Dear Authors,
Thanks for the revision and from my side, Congratulations!
Reviewer 2 Report
Comments and Suggestions for Authors
This can be published in its present form.